# GROUP EQUIVARIANT CONDITIONAL NEURAL PROCESSES

**Makoto Kawano**
The University of Tokyo
Tokyo, Japan
kawano@weblab.t.u-tokyo.ac.jp

**Wataru Kumagai**
The University of Tokyo, RIKEN AIP
Tokyo, Japan
kumagai@weblab.t.u-tokyo.ac.jp

**Akiyoshi Sannai**
RIKEN AIP
Tokyo, Japan
akiyoshi.sannai@riken.jp

**Yusuke Iwasawa & Yutaka Matsuo**
The University of Tokyo
Tokyo, Japan
{iwasawa, matsuo}@weblab.t.u-tokyo.ac.jp

## ABSTRACT

We present the group equivariant conditional neural process (EquivCNP), a meta-learning method with permutation invariance in a data set as in conventional conditional neural processes (CNPs), and it also has transformation equivariance in data space. Incorporating group equivariance, such as rotation and scaling equivariance, provides a way to consider the symmetry of real-world data. We give a decomposition theorem for permutation-invariant and group-equivariant maps, which leads us to construct EquivCNPs with an infinite-dimensional latent space to handle group symmetries. In this paper, we build architecture using Lie group convolutional layers for practical implementation. We show that EquivCNP with translation equivariance achieves comparable performance to conventional CNPs in a 1D regression task. Moreover, we demonstrate that incorporating an appropriate Lie group equivariance, EquivCNP is capable of zero-shot generalization for an image-completion task by selecting an appropriate Lie group equivariance.

## 1 INTRODUCTION

Data symmetry has played a significant role in the deep neural networks. In particular, a convolutional neural network, which play an important part in the recent achievements of deep neural networks, has translation equivariance that preserves the symmetry of the translation group. From the same point of view, many studies have aimed to incorporate various group symmetries into neural networks, especially convolutional operation (Cohen et al., 2019; Defferrard et al., 2019; Finzi et al., 2020). As example applications, to solve the dynamics modeling problems, some works have introduced Hamiltonian dynamics (Greydanus et al., 2019; Toth et al., 2019; Zhong et al., 2019). Similarly, Quessard et al. (2020) estimated the action of the group by assuming the symmetry in the latent space inferred by the neural network. Incorporating the data structure (symmetries) into the models as inductive bias, can reduce the model complexity and improve model generalization.

In terms of inductive bias, meta-learning, or learning to learn, provides a way to select an inductive bias from data. Meta-learning use past experiences to adapt quickly to a new task $\mathcal{T} \sim p(\mathcal{T})$ sampled from some task distribution $p(\mathcal{T})$. Especially in supervised meta-learning, a task is described as predicting a set of unlabeled data (target points) given a set of labeled data (context points). Various works have proposed the use of supervised meta-learning from different perspectives (Andrychowicz et al., 2016; Ravi & Larochelle, 2016; Finn et al., 2017; Snell et al., 2017; Santoro et al., 2016; Rusu et al., 2018). In this study, we are interested in neural processes (NPs) (Garnelo et al., 2018a;b), which are meta-learning models that have encoder-decoder architecture (Xu et al., 2019). The encoder is a permutation-invariant function on the context points that maps the contexts into a latent representation. The decoder is a function that produces the conditional predictive distribution of targets given the latent representation. The objective of NPs is to learn the encoder and the decoder, so that the predictive model generalizes well to new tasks by observing some points of the tasks. To achieve the

objective, an NP is required to learn the shared information between the training tasks $\mathcal{T}, \mathcal{T}' \sim p(\mathcal{T})$: the data knowledge Lemke et al. (2015). Each task $\mathcal{T}$ is represented by one dataset, and multiple datasets are provided for training NPs to tackle a meta-task. For example, we consider a meta-task that completing the pixels that are missing in a given image. Often, images are taken by the same condition in each dataset, respectively. While the datasets contain identical subjects of images (e.g., cars or apples), the size and angle of the subjects in the image may be different; the datasets have group symmetry, such as scaling and rotation. Therefore, it is expected that pre-constraining NPs to have group equivariance improves the performance of the NPs at those datasets.

In this paper, we investigate the group equivalence of NPs. Specifically, we try to answer the following two questions, (1) can NPs represent equivariant functions? (2) can we explicitly induce the group equivariance into NPs? In order to answer the questions, we introduce a new family of NPs, EquivCNP, and show that EquivCNP is a permutation-invariant and group-equivariant function theoretically and empirically. Most relevant to EquivCNP, ConvCNP (Gordon et al., 2019) shows that using general convolution operation leads to the translation equivariance theoretically and experimentally; however it does not consider incorporation of other groups. First, we introduce the decomposition theorem for permutation-invariant and group-equivariant maps. The theorem suggests that the encoder maps the context points into a latent variable, which is a functional representation, in order to preserve the data symmetry. Thereafter, we construct EquivCNP by following the theorem. In this study, we adopt LieConv (Finzi et al., 2020) to construct EquivCNP for practical implementation. We tackle a 1D synthetic regression task (Garnelo et al., 2018a;b; Kim et al., 2019; Gordon et al., 2019) to show that EquivCNP with translation equivariance is comparable to conventional NPs. Furthermore, we design a 2D image completion task to investigate the potential of EquivCNP with several group equivariances. As a result, we demonstrate that EquivCNP enables zero-shot generalization by incorporating not translation, but scaling equivariance.

## 2    Related Work

### 2.1    Neural Networks with Group Equivariance

Our works build upon the recent advances in group equivariant convolutional operation incorporated into deep neural networks. The first approach is group convolution introduced in (Cohen & Welling, 2016), where standard convolutional kernels are used and their transformation or the output transformation is performed with respect to the group. This group convolution induces exact equivariance, but only to the action of discrete groups. In contrast, for exact equivariance to continuous groups, some works employ harmonic analysis so as to find the basis of equivariant functions, and then parameterize convolutional kernels in the basis (Weiler & Cesa, 2019). Although this approach can be applied to any type of general data (Anderson et al., 2019; Weiler & Cesa, 2019), it is limited to local application to compact, unimodular groups. To address these issues, LieConv (Finzi et al., 2020) and other works (Huang et al., 2017; Bekkers, 2019) use Lie groups. Our EquivCNP chooses LieConv to manage group equivariance for simplicity of the implementation.

There are several works that study deep neural networks using data symmetry. In some works, in order to solve machine learning problems such as sequence prediction or reinforcement learning, neural networks attempt to learn a data symmetry of physical systems from noisy observations directly (Greydanus et al., 2019; Toth et al., 2019; Zhong et al., 2019; Sanchez-Gonzalez et al., 2019). While both these studies and EquivCNP can handle data symmetries, EquivCNP is not limited to specific domains such as physics.

Furthermore, Quessard et al. (2020) let the latent space into which neural networks map data, have group equivariance, and estimated the parameters of data symmetries. In terms of using group equivariance in the latent space, EquivCNP is similar to this study but differs from being able to use various group equivariance.

### 2.2    Family of neural processes

NPs (Garnelo et al., 2018a;b) are deep generative models for regression functions that map an input $x_i \in \mathbb{R}^{d_x}$ into an output $y_i \in \mathbb{R}^{d_y}$. In particular, given an arbitrary number of observed data points $(x_C, y_C) := \{(x_i, y_i)\}_{i=1}^{C}$, NPs model the conditional distribution of the target value $y_T$ at some new,

unobserved target data point $x_T$, where $(x_T, y_T) := \{(x_j, y_j)\}_{j=1}^T$. Fundamentally, there are two NP variants: deterministic and probabilistic. Deterministic NPs (Garnelo et al., 2018a), known as conditional NPs (CNPs), model the conditional distribution as:

$$p(y_T|x_T, x_C, y_C) := p(y_T|x_T, r_C),$$

where $r$ represents a function that maps data sets $(x_C, y_C)$ into a finite-dimensional vector space in a permutation-invariant way and $r_C := r(x_C, y_C) \in \mathbb{R}^d$ is the feature vector. The function $r$ can be implemented by DeepSets (Zaheer et al., 2017). The likelihood $p(y_T|x_T, r_C)$ is modeled by Gaussian distribution factorized across the targets $(x_j, y_j)$ with mean and variance of prediction $\{(x_j, y_j)\}_{j=1}^T$ by passing inputs $r_C$ and $x_j$ through the MLP. The CNP is trained by maximizing the likelihood.

Probabilistic NPs include a latent variable $z$. The NP infers $q(z|r_C)$ given an input $r_C$ using the reparametrization trick (Kingma & Welling, 2013) and models such a conditional distribution as:

$$p(y_T|x_T, x_C, y_C) := \int p(y_T|x_T, r_C, z)q(z|r_C)dz$$

and it is trained by maximizing an ELBO: $\mathcal{L}(\phi, \theta) = \mathbb{E}_{z \sim q_\phi(z|x_T, y_T)}[\log p_\theta(y_T|x_T)] - KL[q_\phi(z|x_T, y_T)\|p_\theta(z|x_C, y_C)]$.

NPs have various useful properties: i) Scalability: the computational cost of NPs scales as $\mathcal{O}(n + m)$ with respect to $n$ contexts and $m$ targets of data, ii) Flexibility: NPs can define a conditional distribution of an arbitrary number of target points, conditioning an arbitrary number of observations, iii) Permutation invariance: the encoder of NPs uses Deepsets (Zaheer et al., 2017) to make the target prediction permutation invariant. Thanks to these properties, Galashov et al. (2019) replace Gaussian processes in Bayesian optimization, contextual multi-armed bandit, and Sim2Real tasks.

While there are many NP variants (Kim et al., 2019; Louizos et al., 2019; Xu et al., 2019) to improve the performance of NPs, those do not take group equivariance into account yet. The most similar to EquivCNP, ConvCNP (Gordon et al., 2019) incorporated only translation equivariance. In contrast, EquivCNP can incorporate not only translation but also other groups such as rotation and scaling.

## 3 DECOMPOSITION THEOREM

In this section, we consider group convolution. We first prepare some definition and teminology. Let $\mathcal{X}$ and $\mathcal{Y} \subset \mathbb{R}$ be the input space and output space, respectively. We define $\mathcal{Z}_M = (\mathcal{X} \times \mathcal{Y})^M$ as a collection of $M$ input-output pairs, $\mathcal{Z}_{\leq M} = \bigcup_{n=1}^M \mathcal{Z}_n$ as the collection of at most $M$ pairs, and $\mathcal{Z} = \bigcup_{m=1}^\infty \mathcal{Z}_m$ as the collection of finitely many pairs. Let $[n] = \{1, \ldots, n\}$ for $n \in \mathbb{N}$, and let $\mathbb{S}_n$ be the permutation group on $[n]$. The action of $\mathbb{S}_n$ on $\mathcal{Z}_n$ is defined as

$$\pi Z_n := ((\boldsymbol{x}_{\pi^{-1}(1)}, \boldsymbol{y}_{\pi^{-1}(1)}), \ldots, (\boldsymbol{x}_{\pi^{-1}(n)}, \boldsymbol{y}_{\pi^{-1}(n)})),$$

where $\pi \in \mathbb{S}_n$ and $Z_n \in \mathcal{Z}_n$. We define the multiplicity of $Z_n = ((\boldsymbol{x}_1, \boldsymbol{y}_1), \ldots, (\boldsymbol{x}_n, \boldsymbol{y}_n)) \in \mathcal{Z}_n$ by

$$\text{mult}(Z_n) := \sup \{|\{i \in [n] : \boldsymbol{x}_i = \hat{\boldsymbol{x}}\}| : \hat{\boldsymbol{x}} = \boldsymbol{x}_1, \ldots, \boldsymbol{x}_n\}$$

and the multiplicity of $\mathcal{Z}' \subseteq \mathcal{Z}$ by $\text{mult}(\mathcal{Z}') := \sup_{Z_n \in \mathcal{Z}'} \text{mult}(Z_n)$. Then, a collection $\mathcal{Z}' \subseteq \mathcal{Z}$ is said to have multiplicity $K$ if $\text{mult}(\mathcal{Z}') = K$.

Mathematically, symmetry is described in terms of group action. The following group equivariant maps represent to preserve the symmetry in data.

**Definition 1** (Group Equivariance and Invariance). *Suppose that a group $G$ acts on sets $\mathcal{S}$ and $\mathcal{S}'$. Then, a map $\Phi : \mathcal{S} \to \mathcal{S}'$ is called $G$-equivariant when $\Phi(g \cdot s) = g \cdot \Phi(s)$ holds for arbitrary $g \in G$ and $s \in \mathcal{S}$. In particular, when $G$ acts on $\mathcal{S}'$ trivially (i.e., $g \cdot s' = s'$ for $g \in G$ and $s' \in \mathcal{S}'$), the $G$-equivariant map is said to be $G$-invariant: $\Phi(g \cdot s) = \Phi(s)$.*

Then, we can derive the following theorem, which decompose a permutation-invariant and group equivariant function into two tractable functions. Note that this theorem has been proved by Gordon et al. (2019) when $G$ is a translation group.

**Theorem 2** (Decomposition Theorem). *Let $G$ be a group. Let $\mathcal{Z}'_{\leq M} \subseteq (\mathcal{X} \times \mathcal{Y})_{\leq M}$ be topologically closed, permutation-invariant and $G$-invariant with multiplicity $K$. For a function $\Phi : \mathcal{Z}'_{\leq M} \to C_b(\mathcal{X}, \mathcal{Y})$, the following conditions are equivalent:*

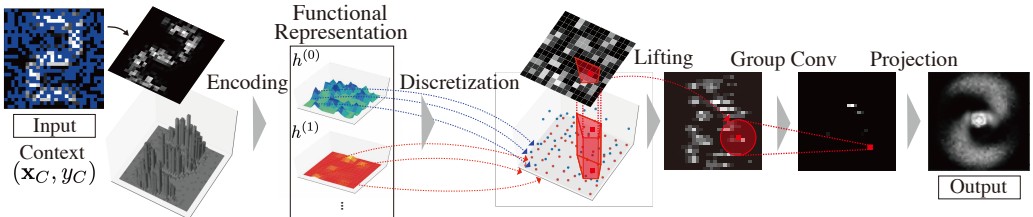

Figure 1: Overview of EquivCNP.

---

**Algorithm 1** Prediction of Group Equivariant Conditional Neural Process

---

**Input:** $\rho =$ LieConv, RBF kernel $\psi$, context $\{\boldsymbol{x}_i, y_i\}_{i=1}^N$, target $\{\boldsymbol{x}_j^*\}_{j=1}^M$

  lower, upper $\leftarrow$ range$((\boldsymbol{x}_i)_{i=1}^N \cup (\boldsymbol{x}_j)_{j=1}^M)$

  $(\boldsymbol{t}_k)_{k=1}^T \leftarrow$ uniform_grid(lower, upper; $\gamma$)

  // Encoding the context information into representation $\boldsymbol{h}$(i.e. Encoder)

  $\boldsymbol{h} \leftarrow \sum_{i=1}^N \phi_{K+1}(y_i)\psi([\boldsymbol{x}_j^*, \boldsymbol{t}_k], \boldsymbol{x}_i)\,\boldsymbol{h}$.

  $(\boldsymbol{\mu}_j, \boldsymbol{\Sigma}_j)^\top = \text{LieConvNet}(\boldsymbol{h})(\boldsymbol{x}_j^*)$ // Decoder

**Output:** $\{(\boldsymbol{\mu}_j, \boldsymbol{\Sigma}_j)\}_{j=1}^M$

---

  *(I) $\Phi$ is continuous, permutation-invariant and $G$-equivariant.*

  *(II) There exist a function space $\mathcal{H}$ and a continuous $G$-equivariant function $\rho : \mathcal{H} \to C_b(\mathcal{X}, \mathcal{Y})$ and a continuous $G$-invariant interpolating kernel $\psi : \mathcal{X}^2 \to \mathbb{R}$ such that*

$$\Phi(Z) = \rho \left( \sum_{i=1}^m \phi_{K+1}(y_i)\,\psi_{\boldsymbol{x}_i} \right)$$

  *where $\phi_{K+1} : \mathcal{Y} \to \mathbb{R}^{K+1}$ is defined by $\phi_{K+1}(y) := [1, y, y^2, \ldots, y^K]^\top$.*

Thanks to the Theorem 2, we can construct the permutation-invariant and group-equivariant NPs whose form of encoder and decoder is determined. In this paper, we call $\Phi$ as EquivDeepSet.

## 4 Group Equivariant Conditional Neural Processes

In this section, we represent EquivCNP that is a permutation-invariant and group-equivariant map. EquivCNP models the same conditional distribution as well as CNPs:

$$p(\boldsymbol{Y}_T | \boldsymbol{X}_T, \mathcal{D}_C) = \prod_{n=1}^N p(\boldsymbol{y}_n | \Phi_{\boldsymbol{\theta}}(\mathcal{D}_C)(\boldsymbol{x}_n))$$

$$= \prod_{n=1}^N \mathcal{N}(\boldsymbol{y}_n; \boldsymbol{\mu}_n, \boldsymbol{\Sigma}_n) \text{ with } (\boldsymbol{\mu}_n, \boldsymbol{\Sigma}_n) = \Phi_{\boldsymbol{\theta}}(\mathcal{D}_C)(\boldsymbol{x}_n)$$

where $\mathcal{N}$ denotes the density function of a normal distribution, $\mathcal{D}_C = (\boldsymbol{X}_C, \boldsymbol{Y}_C) = \{(x_c, y_c)\}_{i=1}^C$ is the observed context data and $\phi$ is a EquivDeepSet. The important components of EquivCNP to be determined are $\rho$, $\phi$, and $\psi$. The algorithm is represented in Algorithm 1.

To describe in more detail, first, Section 4.1 introduce the definition of group convolution, and then Section 4.2 explains LieConv (Finzi et al., 2020) used for EquivCNP to implement group convolution. Finally, we describe the architecture of proposed EquivCNP in Section 4.3.

### 4.1 Group Convolution

When $\mathcal{X}$ is a homogenous space of a group $G$, the lift of $x \in \mathcal{X}$ is the element of group $G$ that transfers a fixed origin $o$ to $x$ : Lift$(x) = \{u \in G : uo = x\}$. That is, each pair of coordinates and

features is lifted into $K$ elements[1] : $\{(x_i, f_i)\}_{i=1}^N \rightarrow \{(u_{ik}, f_i)\}_{i=1,k=1}^{N,K}$. When the group action is transitive, the space on which it acts on is a homogenous space. More generally, however, the action is not transitive, and the total space contains an infinite number of orbits. Consider a quotient space $Q = \mathcal{X}/G$, which consists of orbits of $G$ in $\mathcal{X}$. Then each element $q \in Q$ is a homogenous space of $G$. Because many equivariant maps use this information, the total space should be $G \times \mathcal{X}/G$, not $G$. Hence, $x \in \mathcal{X}$ is lifted to the pair $(u, q)$, where $u \in G$ and $q \in Q$.

Group convolution is a generalization of convolution by translation, which is used in images, etc., to other groups.

**Definition 3** (Group Convolution (Kondor & Trivedi, 2018; Cohen et al., 2019)). *Let $g, f \colon G \times Q \rightarrow \mathbb{R}$ be functions, and let $\mu(\cdot)$ be a Haar measure on $G$. For any $u \in G$, the convolution of $f$ by $g$ is defined as*

$$h(u, q) = \int_{G \times Q} g(v^{-1}u, q, q')f(v, q')d\mu(v)dq'.$$

By the definition, we can verify that the group convolution is $G$-equivariant. Moreover, Cohen et al. (2019) recently showed that a $G$-equivariant linear map is represented by group convolution when the action of a group is transitive.

## 4.2 LOCAL GROUP CONVOLUTION

In this study, we used LieConv as a group convolution (Finzi et al., 2020). LieConv is a convolution that can handle Lie groups in group convolutions. LieConv acts on a pair $(x_i, f_i)_{i=1}^N$ of coordinates $x_i \in \mathcal{X}$ and values $f_i \in V$ in vector space $V$. First, input data $x_i$ is transformed (lifted) into group elements $u_i$ and orbits $q_i$. Next, we define the convolution range based on the invariant (pseudo) distance in the group, and convolve it using a kernel parameterized by a neural network.

What is important for inductive bias and computational efficiency in convolution is that the range of convolutions is local; that is, if the distance between $u_i$ and $u_j$ is larger than $r$, $g_\theta(u_i, u_j) = 0$. First, we define distance in the Lie group to deal with locality in the matrix group[2]:

$$d(u, v) := \| \log(u^{-1}v) \|_F,$$

where $\log$ denotes the matrix logarithm, and $F$ denotes the Frobenius norm. Because $d(wu, wv) = \| \log(u^{-1}w^{-1}wv) \|_F = d(u, v)$ holds, this function is left-invariant and is a pseudo-distance.[3]

To further account for orbit $q$, we extend the distance to $d((u_i, q_i), (v_j, q_j))^2 = d(u_i, v_j)^2 + \alpha d_{\mathcal{O}}(q_i, q_j)^2$, where $d_{\mathcal{O}}(q_i, q_j) := \inf_{x_i \in q_i, x_j \in q_j} d_{\mathcal{X}}(x_i, x_j)$ and $d_{\mathcal{X}}$ is the distance on $\mathcal{X}$. It is not necessarily invariant to the transformation in $q$.

Based on this distance, the neighborhood is $nbhd(u) = \{v, q | d((u_i, q_i), (v_i, q_i)) < r\}$. The radius $r$ should be adjusted appropriately from the ratio of the range of convolutions to the total input, because the appropriate value is difficult to determine depending on the group treated. Therefore, the Lie group convolution is

$$h(u, q) = \int_{v, q' \in \text{nbhd(u)}} g_\theta(v^{-1}u, q, q')f(v, q')d\mu(v)dq'.$$

Radius $r$ of the neighborhood corresponds to the inverse of the density channel $h^{(0)}$ in Gordon et al. (2019).

**Discrete Approximation**. Given a lifted input data point $\{(v_j, q_j)_{j=1}^N\}$ and a function value $f_j = f(v_j, q_j)$ at each point, we need to select a target $\{(u_i, q_i)_{i=1}^N\}$ to convolve so that we can approximate the integral of the equation. Because the convolutional range is limited by $\text{nbhd}(u)$, LieConv can approximate the integrals by the Monte Carlo method：

$$h(u, q) = (g \hat{*} f)(u, q) = \frac{1}{n} \sum_{v_j, q_j' \in \text{nbhd}(u,q)} g(v_j^{-1}u, q, q_j')f(v_j, q_j')$$

---

[1] $K$ is a hyperparameter and we randomly pick $K$ elements $\{u_{ik}\}_{k=1}^K$ in the orbit corresponding to $x_i$.

[2] We assume that we have a finite-dimensional representation.

[3] This is because the triangle inequality is not satisfied.

The classical convolutional filter kernel $g(\cdot)$ is only valid for discrete values and is not available for continuous group elements. Therefore, pointconv/Lieconv uses a multilayered neural network $g_\theta$ as a convolutional kernel. However, because neural networks are good at computation in Euclidean space, and input $G$ is not a vector space, we let $g_\theta$ be a map in the Lie algebra $\mathfrak{g}$. Therefore, we use Lie groups and logarithmic maps exist in each element of the group. That is, let $g_\theta(u) = (g \circ \exp)_\theta(\log u)$, and parameterize $\tilde{g}_\theta = (g \circ \exp)_\theta$ by MLP. We use $\tilde{g}_\theta \colon \mathfrak{g} \to \mathbb{R}^{c_{out} \times c_{in}}$. Therefore, the convolution of the equation is

$$h_i = \frac{1}{n_i} \sum_{j \in \text{nbhd}(i)} \tilde{g}_\theta \left( \log \left( v_j^{-1} u_i \right), q_i, q_j \right) f_j.$$

Here, the input to the MLP is $a_{ij} = \text{Concat}\left( [\log(v_j^{-1} u_i), q_i, q_j)] \right)$.

### 4.3 IMPLEMENTATION

First, we explain the form of $\phi$. Because most real-world data have a single output per one input location, we treat the multiplicity of $\mathcal{D}_C$ as one, $K = 1$, and define $\phi(y) = \begin{bmatrix} 1 & y \end{bmatrix}^\top$ based on (Zaheer et al., 2017). The first dimension of output $\phi_i$ indicates whether the data located at $x_i$ is observed, so that the model can distinguish between the observed data, and the unobserved data whose value is zero ($y_i = 0$).

Then, we describe the form of $\psi$. Following our Theorem 2, $\psi$ is required to be stationary, non-negative, and a positive–definite kernel. For EquivCNP, we change $\psi$ depending on whether the input data is continuous or discrete. With continuous input data (e.g. 1D regression), we use RBF kernels for $\psi$. An RBF kernel has a learnable bandwidth parameter and scale parameter and is optimized with EquivCNP. A functional representation $E(Z)$ is made up by multiplying the kernel $\psi$ with $\phi$. On the other hand, when the inputs are discrete (e.g. images), we use not an RBF kernel but LieConv.

Finally, we explain the form of $\rho$. With our Theorem2, because $\rho$ needs to be a continuous group equivariant map between function spaces, we use LieConv for $\rho$. In this study, under the hypothesis of separability (Kaiser et al., 2017), we implemented separable LieConv in the spatial and channel directions, to improve the efficiency of computational processing. The details are given in the Appendix B. EquivCNP requires to compute the convolution of $E(Z)$. However, since $E(Z)$ itself is a functional representation, it cannot be computed in computers as it is. To address this issue, we discretize $E(Z)$ over the range of context and target points. We space the lattice points $(t_i)_{i=1}^n \subseteq \mathcal{X}$ on a uniform grid over a hypercube covering both the context and target points. Because the conventional convolution that is used in ConvCNP requires discrete lattice input space to operate on and produces discrete outputs, we need to back the outputs to continuous functions $\mathcal{X} \to \mathcal{Y}$. While ConvCNP regards the outputs as weights for evenly-spaced basis functions (i.e., RBF kernel), LieConv does not require the input location to be lattice and can produce continuous functions output directly. Note that the algorithm of EquivCNP can be the same as ConvCNP; it can also use evenly-spaced basis functions. The obtained functions are used to output the Gaussian predictive mean and the variance at the given target points. We can evaluate EquivCNP by log-likelihood using the mean and variance.

## 5 EXPERIMENT

To investigate the potential of EquivCNP, we constructed three questions: 1) Is EquivCNP comparable to conventional NPs such as ConvCNP? and 2) Can EquivCNP have group equivariance in addition to translation equivariance and 3) does it preserve the symmetries? To compare fairly with ConvCNP, the architecture of EquivCNP follows that of ConvCNP; details are given in the Appendix C.

### 5.1 1D SYNTHETIC REGRESSION TASK

To answer the first question, we tackle the 1D synthetic regression task as has been done in other papers (Garnelo et al., 2018a;b; Kim et al., 2019). At each iteration, a function $f$ is sampled from a given function distribution, then, some of the context $\mathcal{D}_C$ and target $\mathcal{D}_T$ points are sampled from function $f$. In this experiment, we selected the Gaussian process with RBF kernel, Matern–$\frac{5}{2}$ and periodic kernel for the function distribution. We chose translation equivariance $T(1)$ to incorporate

Table 1: Log-likelihood of synthetic 1-dimensional regression

| Model | RBF | Matern | Periodic |
|---|---|---|---|
| Oracle GP | $3.9335 \pm 0.5512$ | $3.7676 \pm 0.3542$ | $1.2194 \pm 5.6685$ |
| CNP (Garnelo et al., 2018a) | $-1.7468 \pm 1.5415$ | $-1.7808 \pm 1.3124$ | $-1.0034 \pm 0.5174$ |
| ConvCNP (Gordon et al., 2019) | $1.3271 \pm 1.0324$ | $0.8189 \pm 0.9366$ | $-0.4787 \pm 0.5448$ |
| EquivCNP (ours) | $1.2930 \pm 1.0113$ | $0.6616 \pm 0.6728$ | $-0.4037 \pm 0.4968$ |

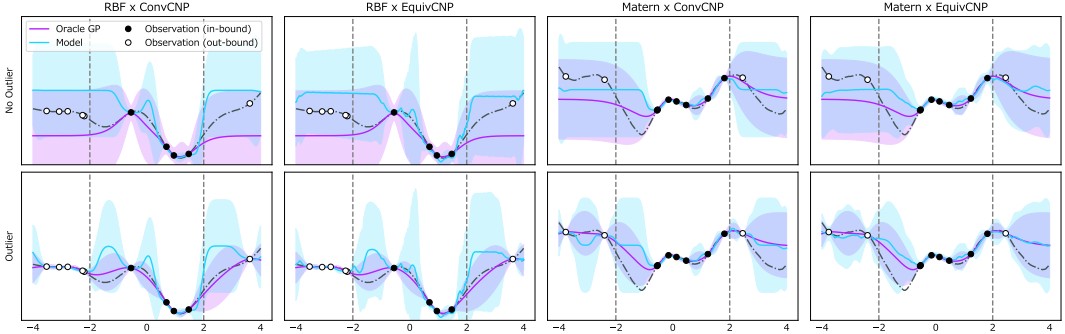

Figure 2: Predictive mean and variance of ConvCNP and EquivCNP. The first two columns show the prediction of the models trained on the RBF kernel and the last two columns show the prediction of the model trained on the Matern$-\frac{5}{2}$ kernel. The target function and sampled data points are the same between the top row and bottom row except for the context. At the top row, the context is within the vertical dash line that is sampled from the same range during the training (black circle). In the bottom row, the new context located out of the training range (white circle) is appended.

into EquivCNP. We compared EquivCNP with GP (as an oracle), with CNP (Garnelo et al., 2018a) as a baseline, and with ConvCNP.

Table 1 shows the log–likelihood means and standard deviations of 1000 tasks. In this task, both contexts and targets are sampled from the range $[-2, 2]$. From Table 1, we can see that EquivCNP with translation equivariance is comparable to ConvCNP throughout all GP curve datasets. That is, EquivCNP has the model capacity to learn the functions as well as ConvCNP.

We also conducted the extrapolation regression proposed in (Gordon et al., 2019) as shown in Figure 2. The first two columns show the models trained on an RBF kernel and the last two columns on a Matern$-\frac{5}{2}$ kernel. The top row shows the predictive distribution when the observation is given within the same training region; the bottom row for the observation is not only the training region but also the extrapolation region: $[-4, 4]$. As a result, EquiveCNP can generalize to the observed data whose range is not included during training. This result was expected because Gordon et al. (2019) has mentioned that translation equivariance enables the models to adapt to this setting.

## 5.2 2D IMAGE-COMPLETION TASK

An image-completion task aims to investigate that EquivCNP can complete the images when it is given an appropriate group equivariance. The image-completion task can be regarded as a regression task that predicts the value of $y_i^*$ at the 2D image coordinates $x_i^*$, given the observed pixels $\mathcal{D}_C = \{(x_n, y_n)\}_{n=1}^{N}$ ($\in \mathbb{R}^3$ for the colored image input, and $\in \mathbb{R}$ for the grayscale image input). The framework of the image completion can apply not only to the images but also to other real-world applications, such as predicting spatial data (Takeuchi et al., 2018).

To evaluate the effect of EquivCNP with a specific group equivariance, we introduce a new dataset digital clock digits as shown in Figure 3. Since previous works use the MNIST dataset for image completion, we also conduct the image completion task with rotated-MNIST. However, we cannot find a significant difference between the group equivariance models (the result of rotated-MNIST is depicted in Appendix E). We think that this happens because (1) original MNIST contains various

Table 2: Log-likelihood of 2D image-completion task

| Group | Log–likelihood |
|---|---|
| $T(2)$ | $1.0998 \pm 0.4115$ |
| $SO(2)$ | $-2.4275 \pm 6.8856$ |
| $R_{>0} \times SO(2)$ | $\mathbf{1.8398 \pm 0.5368}$ |
| $SE(2)$ | $1.1655 \pm 0.5420$ |

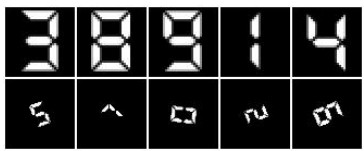

Figure 3: The example of training data (top) and test data (bottom).

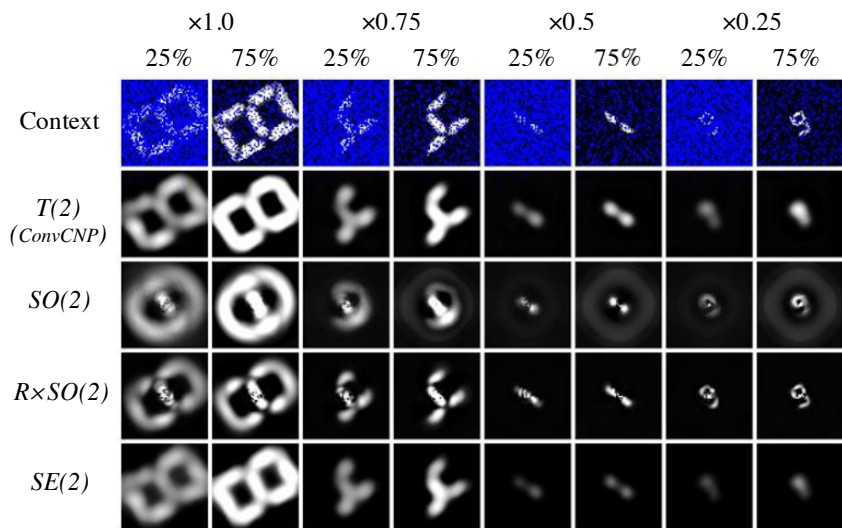

Figure 4: Image-completion task results. The top row shows the given observation and the other rows show the mean of the conditional distribution predicted by EquivCNP with the specific group equivariance: $T(2)$, $SO(2)$, $R_{>0} \times SO(2)$, and $SE(2)$. Two of each column shows the same image, and the difference between two columns is the percentage of context random sampling: $25\%$ and $75\%$. When the size of digits is the same as that of the training set (i.e. not scaling but rotation equals $SO(2)$ symmetry), $T(2)$ and $SE(2)$ have a good quality, but when the size of digits is smaller than that of training set, $R_{>0} \times SO(2)$ has a good performance.

data symmeries including translation, scaling, and rotation, and (2) we cannot specify them precisely. Thus, we provide digital clock digits dataset anew.

In this experiment, we used four kinds of group equivariance; translation group $T(2)$, the 2D rotation group $SO(2)$, the translation and rotation group $SE(2)$, and the rotation-scale group $R_{>0} \times SO(2)$. The size of the images is $64 \times 64$ pixels, and the numbers are in the center with the same vertical length. For the test data, we transform the images by scaling within $[0.15, 0.5]$ and rotating within $[-90°, +90°]$. Image completion with our digits data becomes an extrapolation task in that the test data is never seen during training, though the number shapes are the same in both sets.

The log–likelihood of image completion by EquivCNP with the group equivariance is reported in Table 2. The mean and standard deviation of the log–likelihood is calculated over 1000 tasks (i.e. evaluating the digit transformed in 100 times respectively). As a result, EquivCNP with $R_{>0} \times SO(2)$ performed better than other group equivarinace. On the other hand, the model with $SO(2)$ had the worst performance. This might happen because the $SO(2)$ is not able to generalize EquivCNP to scaling. In fact, the log–likelihood of $SE(2)$, which is the group equivariance combining translation $T(2)$ and rotation $SO(2)$, is not improved than that of $T(2)$.

Figure 4 shows the qualitative result of image completion by EquivCNP with each group equivariance. We demonstrate that EquivCNP was able to predict digits smaller than the training digits[4]. While $T(2)$ completes the images most clearly when the sizes of digits and the number of observations

---

[4]When the scaling is $\times 1.0$, it equals to $SO(2)$ symmetry.

are large, other groups also complete the images. The smaller the size of digits is compared to the training digits, the worse the quality of $T(2)$ completion becomes, and $R_{>0} \times SO(2)$ completes the digits more clearly. This is because the convolution region of $T(2)$ is invariant to the location, while that of $R_{>0} \times SO(2)$ is adaptive to the location. As a result, for the images transformed by scaling, we can see that EquivCNP with $R_{>0} \times SO(2)$ preserved scaling group equivariance.

## 6 DISCUSSION

We presented a new neural process, EquivCNP, that uses the group equivariant adopted from LieConv. Given a specific group equivariance, such as translation and rotation as inductive bias, EquivCNP has a good performance at regression tasks. This is because the kernel size changes depending on the specific equivariance. Real–world applications, such as robot learning tasks (e.g. using hand-eye camera) will be left as future work. We also hope EquivCNPs will help in learning group equivariance (Quessard et al., 2020) by data–driven approaches for future research.

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

SUPPLEMENTARY MATERIAL

A. PROOF OF THEOREM 2

First, we prove that (II) implies (I). We define the action of $G$ on the set of univariate maps $f : \mathcal{X} \to \mathbb{R}$ by

$$(g \cdot f)(\boldsymbol{x}) := f(g^{-1} \cdot \boldsymbol{x}).$$

and define the action of $G$ on the set of bivariate maps $\psi : \mathcal{X}^2 \to \mathbb{R}$ by

$$(g \cdot \psi)(\boldsymbol{x}, \boldsymbol{x}') := \psi(g^{-1} \cdot \boldsymbol{x}, g^{-1} \cdot \boldsymbol{x}').$$

**Lemma 4.** *For a map $\psi : \mathcal{X}^2 \to \mathbb{R}^d$ and a sample $\boldsymbol{x} \in \mathcal{X}$, a sample dependent function $\psi_{\boldsymbol{x}} : \mathcal{X} \to \mathbb{R}^d$ is defined by*

$$\psi_{\boldsymbol{x}}(\boldsymbol{x}') := \psi(\boldsymbol{x}', \boldsymbol{x})$$

*Here, $\psi$ is $G$-invariant if and only if the map $\mathcal{X} \ni \boldsymbol{x} \mapsto \psi_{\boldsymbol{x}} \in \mathsf{Map}(\mathcal{X}, \mathbb{R}^d)$ is $G$-equivariant.*

The above lemma is derived as follows:

$$\psi_{g \cdot \boldsymbol{x}'}(\boldsymbol{x}) = \psi(\boldsymbol{x}, g \cdot \boldsymbol{x}') = \psi(g^{-1} \cdot \boldsymbol{x}, \boldsymbol{x}') = \psi_{\boldsymbol{x}'}(g^{-1} \cdot \boldsymbol{x}) = (g \cdot \psi_{\boldsymbol{x}'})(\boldsymbol{x}).$$

The left hand side represents the action on the sample space and the right hand side does the action on the function space.

When we denote the set of all $G$-equivariant maps from $\mathcal{S}$ to $\mathcal{S}'$ by $\mathsf{Equiv}(\mathcal{S}, \mathcal{S}')$, the above lemma is represented as

$$\mathsf{Inv}(\mathcal{X}^2, \mathbb{R}^d) \cong \mathsf{Equiv}(\mathcal{X}, \mathsf{Map}(\mathcal{X}, \mathbb{R}^d)).$$

Thus, since $\psi : \mathcal{X}^2 \to \mathbb{R}$ is invariant from (II), $\boldsymbol{x} \mapsto \psi_{\boldsymbol{x}}$ is equivariant. Then, for $Z \in \mathcal{Z}'_{\leq M}$, the correspondence $Z \mapsto \sum_{i=1}^m \phi_{K+1}(y_i)\psi_{\boldsymbol{x}_i}$ is also equivariant. Since $\rho$ is equivariant from (II), we obtain (I) because the composition of equivariant maps is equivariant.

Next, we prove that (I) implies (II). We prepare some notations and lemmas in the following. Let $\psi$ be an interpolating continuous kernel that satisfies $\psi(\boldsymbol{x}, \boldsymbol{x}') \geq 0$. Then, for $m \in \mathbb{N}$ and $\mathcal{Z}'_m \subseteq (\mathcal{X} \times \mathcal{Y})^m$, define

$$\mathcal{H}_m(\mathcal{Z}'_m) := \left\{ \sum_{i=1}^m \phi_{K+1}(y_i)\psi(\cdot, \boldsymbol{x}_i) : (\boldsymbol{x}_i, y_i)_{i=1}^m \subseteq \mathcal{Z}'_m \right\} \subseteq \mathcal{H}^{K+1},$$

where $\mathcal{H}^{K+1} = \mathcal{H} \times \cdots \times \mathcal{H}$ is the $(K+1)$-dimensional-vector-valued-function Hilbert space constructed from the RKHS $\mathcal{H}$ for which $\psi$ is a reproducing kernel and endowed with the inner product $\langle f, g \rangle_{\mathcal{H}^{K+1}} = \sum_{i=1}^{K+1} \langle f_i, g_i \rangle_{\mathcal{H}}$, where $\langle \cdot, \cdot \rangle_{\mathcal{H}}$ is the inner product of the RKHS $\mathcal{H}$. When the permutation group $S_m$ acts on a set $(\mathcal{X} \times \mathcal{Y})^m$, the set of equivalence classes of this action is denoted by $(\mathcal{X} \times \mathcal{Y})^m / S_m$. Then, for an element $Z \in (\mathcal{X} \times \mathcal{Y})^m$, the equivalent class of the action is denoted by $[Z]$. Similarly, for a subset $\mathcal{Z}'_m \subset (\mathcal{X} \times \mathcal{Y})^m$, the set of equivalent classes is denoted by $[\mathcal{Z}'_m] := \{[Z] | Z \in \mathcal{Z}'_m\}$. Furthermore, we denote as

$$\left[\mathcal{Z}'_{\leq M}\right] := \bigcup_{m=1}^M [\mathcal{Z}'_m] \quad \text{and} \quad \mathcal{H}_{\leq M} := \bigcup_{m=1}^M \mathcal{H}_m(\mathcal{Z}'_m).$$

Lemma 1 and Lemma 3 in Gordon et al. (2019) provides the following lemma.

**Lemma 5.** *For $m \in \mathbb{N}$, let $\mathcal{Z}'_m \subseteq (\mathcal{X} \times \mathcal{Y})^m$ be a set with multiplicity $K$ and $\psi$ be an interpolating continuous kernel. Then, $(\mathcal{H}_m(\mathcal{Z}'_m))_{m=1}^M$ are pairwise disjoint and the embedding $E$ is injective and continuous:*

$$E : \left[\mathcal{Z}'_{\leq M}\right] \to \mathcal{H}_{\leq M}(\mathcal{Z}'_m), \quad E([Z]) := E_m([Z]) \quad \text{if} \quad [Z] \in [\mathcal{Z}'_m],$$

*where*

$$E_m : [\mathcal{Z}'_m] \to \mathcal{H}_m(\mathcal{Z}'_m), \quad E_m\left([(\boldsymbol{x}_1, y_1), \ldots, (\boldsymbol{x}_m, y_m)]\right) := \sum_{i=1}^m \phi_{K+1}(y_i)\psi(\cdot, \boldsymbol{x}_i)$$

Similarly, Lemma 2 and Lemma 4 in Gordon et al. (2019) provides the following lemma.

**Lemma 6.** *Suppose that $\mathcal{Z}'_M$ is a topologically closed set in $(\mathcal{X} \times \mathcal{Y})^M$ and permutation-invariant, and that $\psi$ satisfies (i) $\psi \geq 0$, (ii) $\psi(x,x) = \sigma^2 > 0$ for any x, and (iii) $\psi(x,x') \rightarrow 0$ as $\|x\| \rightarrow \infty$. Let $\Phi : \left[ \mathcal{Z}'_{<M} \right] \rightarrow C_b(\mathcal{X},\mathcal{Y})$ be a map such that every restriction $\Phi|_{[\mathbf{z}'_m]}$ is continuous. Then, $\Phi \circ E^{-1} : \mathcal{H}_{<M} \rightarrow C_b(\mathcal{X},\mathcal{Y})$ is continuous.*

When a $G$-equivariant function $f$ is injective, $f^{-1}|_{\mathrm{Im}f}$ is also $G$-equivariant on the image of $f$. Denoting $\Phi \circ E^{-1}$ by $\rho$, we can rewrite as $\Phi = \rho \circ E$.

## B. Separable LieConv

In this section, we introduce the separable LieConv that we design and implement for EquivCNP. LieConv (Finzi et al., 2020) is based on PointConv (Wu et al., 2019), which is proposed for point cloud convolution. That is, the lifted inputs are convolved by PointConv. Therefore, we can adopt techniques that are used for general convolution. One of such techniques is separable convolution(Chollet, 2017). Separable convolution consists of depthwise convolution and pointwise convolution (as known as 1 x 1 convolution). The mathematical formulation of normal convolution, the pointwise convolution, and the depthwise convolution is as follow:

$$\mathrm{Conv}(W,y)(i,j) = \sum_{k,l,m}^{K,L,M} W_{(k,l,m)} \cdot y_{(i+k,j+l,m)}$$

$$\mathrm{PointwiseConv}(W,y)(i,j) = \sum_{m}^{M} W_m \cdot y_{(i,j,m)}$$

$$\mathrm{DepthwiseConv}(W,y)(i,j) = \sum_{k=1}^{K,L} W_{(k,l)} \odot y_{(i+k,j+l)}$$

$$\mathrm{SepConv}\left(W_p, W_d, y\right)(i,j) = \mathrm{PointwiseConv}(i,j)\left(W_p, \mathrm{Depthwise\,Conv}\,(i,j)\,(Wd,y)\right)$$

Thanks to the assumption that convolution operation is separable to the spatial direction and the channel direction, the separable convolution provides the way to operate convolution more efficiently than general convolution. Note that the difference of the efficiency between LieConv and separable LieConv is slight; the difference between the matrix production and element-wise product. Following the equation above, we design and implemented separable LieConv. Figure 5 illustrates the processing of (a) normal LieConv and (b) separable LieConv. The memory consumption is also different that the output shape of after the convolutional weights (kernel) is calculated in normal LieConv is $B \times N_{MC} \times C_{mid}$ while that of separable LieConv is $B \times N_{MC} \times C_{in}$.

## C. EquivCNP Architecture

The architecture of EquivCNP is following that of ConvCNP (Gordon et al., 2019), so that we can fairly compare them. It is difficult to determine radius $r$ of LieConv because the radius is varied substantially between the different groups due to the different distance functions. Instead, we parametrized the radius by specifying the average fraction of the total number of convolved elements that would fall into this radius. Therefore, we describe the value of the average fraction instead of kernel size that is described in other papers as usual. Simultaneously, while the conventional convolutional layer has a parameter called *stride* that determines the target elements (pixels) to be convolved, LieConv has a parameter sampling fraction instead of stride to subsample the group elements; sampling fraction is 1.0.

### C.1 1D Synthetic Regression Task

For 1D regression tasks, we use 4-layer LieConv architecture with ReLU activations. The average fraction of those LieConv is $\frac{5}{32}$ and the number of MC sampling is 25. The channels of LieConv are $[16, 32, 16, 8]$. Functional representation $E(Z)$ is concatenated with target point $x_T$, followed by

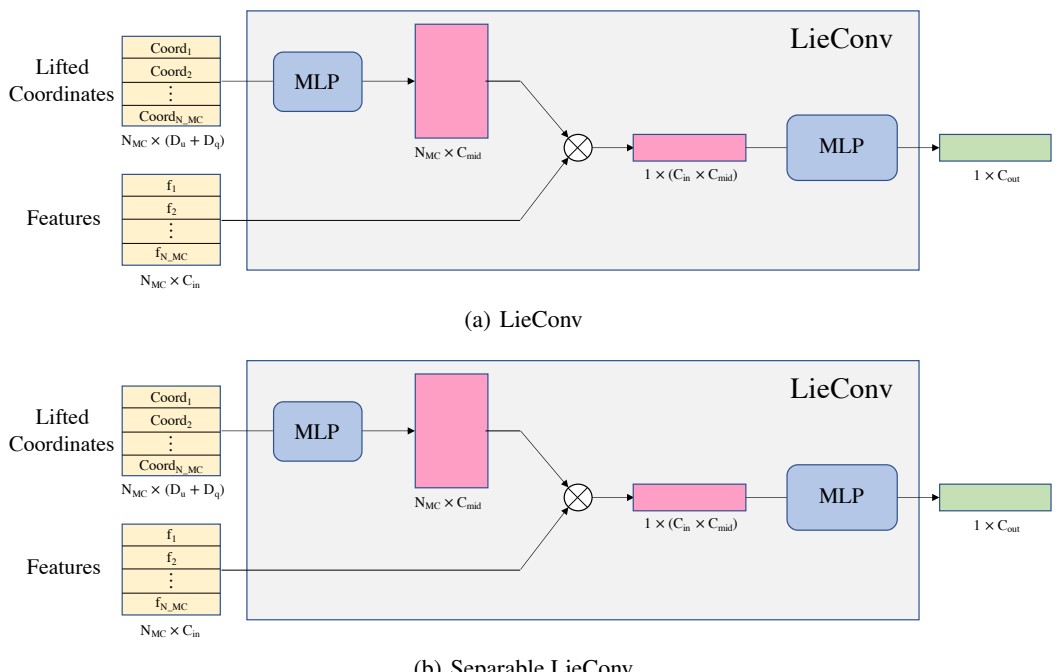

(a) LieConv

(b) Separable LieConv

Figure 5: Separable LieConv. Difference between (a) normal LieConv and (b) separable LieConv is the matrix product $\otimes$ and element-wise product $\odot$.

lifting. After operating convolution to the lifted inputs, we use a softplus activation following the last fully-connected layer (FC) as a standard deviation. Note that the output of EquivCNP, mean and standard deviation, is sliced to get those of $y_T$. The architecture of EquivCNP for a 1D regression task is illustrated in Figure 6.

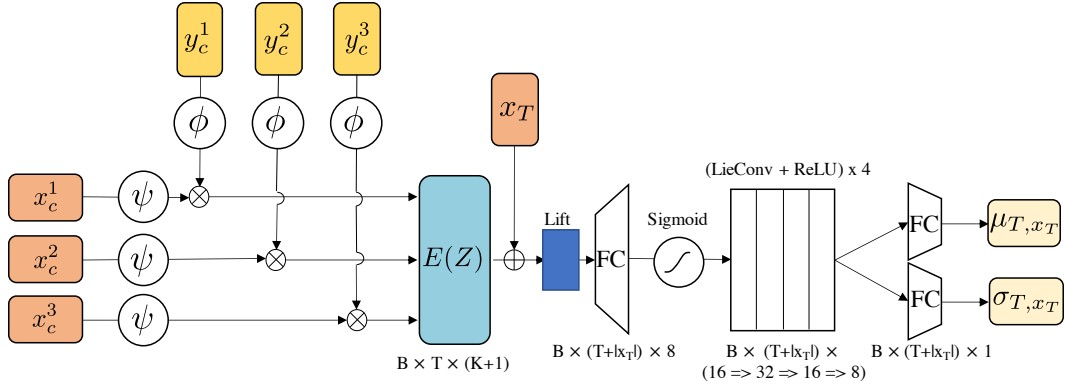

Figure 6: The architecture of EquivCNP for a 1D regression task. $\otimes$ represents dot product and $\oplus$ represents concatenation. $\psi$ is a RBF kernel and $\phi = [y^0, y^1, \ldots, y^K]$.

### C.2 2D IMAGE-COMPLETION TASK

For the 2D image-completion task, we use LieConv $\text{Conv}_\theta$ instead of RBF kernels as $\psi$. The channels of this LieConv is 128, the average fraction is $\frac{1}{10}$, and the number of MC sampling is 121. After the LieConv of $\psi$, we use four residual blocks. Each block is composed by two separable LieConv layers

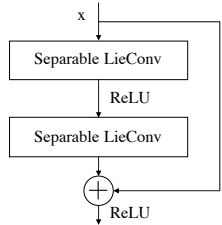

Figure 7: Residual Block

and residual connections as shown in Figure 7. The channel of each residual block is 128, the average fraction is $\frac{1}{15}$, and the number of MC sampling is 81.

We employ the same procedure of ConvCNP (Gordon et al., 2019) for image-completion as follows:

1. Given an input image $I \in \mathbb{R}^{C \times H \times W}$, where $C$ is color channel, $H$ and $W$ represents height and width respectively, sample context points features $:= I \odot M_c$ from bernoulli distribution. $M_c$ means the density as same as we define $\phi$ during 1D regression task.

2. After lifting the inputs, apply a LieConv to both $I \odot M_c$ and $M_c$ to get functional representation: $E(Z) = \text{Conv}_\theta([M_c, I \odot M_c]) \in \mathbb{R}^{(128+128) \times H \times W}$.

3. Then, functional representation $E(Z)$ is passed through one FC followed by four residual blocks: $h = \text{ResBlocks}(\text{FC}(E(Z))) \in \mathbb{R}^{128 \times H \times W}$.

4. Finally, we use one FC to get mean and standard deviation channels and split the output $\in \mathbb{R}^{2C \times H \times W}$ into those statistics.

## D. EXPERIMENT DETAILS

In this section, we describe the experiments in more detail. Code and dataset are available on `https://github.com/makora9143/EquivCNP`.

### D.1 1D SYNTHETIC REGRESSION TASK

The kernels used in Section 5.1 for generating the data via Gaussian Processes are defined as follows:

- RBFKernel:

$$k(x_1, x_2) = \exp\left(-\frac{(x_1 - x_2)^2}{2}\right)$$

- Matern-$\frac{5}{2}$

$$k(x_1, x_2) = \left(1 + \sqrt{5}d + \frac{5}{3}d^2\right)\exp\left(-\sqrt{\frac{5}{2}}d\right) \quad \text{with} \quad d = \|x_1 - x_2\|_2$$

- Periodic

$$k(x_1, x_2) = \exp\left(-2\sin(\pi\|x_1 - x_2\|_2)\right)$$

To train all NPs, the GPs generate the context and target points; the number of context points and target points is random-sampled uniformly from $[3, 50]$ respectively. All NPs were trained for 200 epochs by 256 batches per epoch and the size of each batch is 16, We used Adam optimizer (Kingma & Ba, 2014) with learning rate $10^{-3}$. An architecture of CNP was based on the original code[5]. We visualize the result of periodic kernel regression at Figure 8.

We also demonstrate EquivCNP with the algorithm following that of ConvCNP (Gordon et al., 2019); regarding the output of EquivCNP as weights for evenly-spaced basis functions (i.e. RBF kernel) in Figure 9. The result of predictive distribution is much smoother than the result of our Algorithm 1 though using RBF kernel is redundant.

---

[5]https://github.com/deepmind/neural-processes

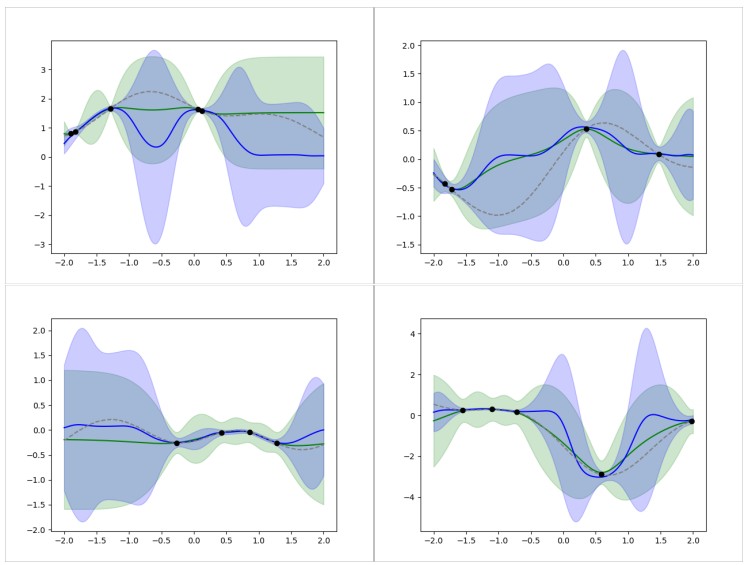

Figure 8: Predictive mean and variance of ConvCNP and EquivCNP at periodic kernels. First two columns show the result without outlier observation and last two columns show the result with outlier observation.

Figure 9: Predictive mean and variance of EquivCNP that using algorithm proposed in (Gordon et al., 2019). Blue line and region represents EquivCNP and green line and region represents Gaussian Process. Each plot shows diffent sampled data. Although the algorithm is redundant compared with our proposed Algorithm 1 due to using RBF kernel to map the output of LieConv back to a continuous function, the result is much smoother than Figure 2 and 8.

## D.2 2D IMAGE-COMPLETION TASK

The original image of the digital clock number is shown in Figure 10. We first inverted in colors of black and white of the image. Then, we cropped the image so that each cropped image contains one digit and resize them to $64 \times 64$. Note that the vertical size of each number is set up to 56, while the horizontal size is not fixed. The values of all pixels are devided by 255 to rescale them to the $[0, 1]$ range.

As we mentioned in Section C.2, the context points are sampled from bernoulli distribution. The parameter of bernoulli distribution, probability $p$ that the value is 1, is determined at a rate of the number uniformly from $\mathcal{U}\left(\frac{n_{\text{total}}}{100}, \frac{n_{\text{total}}}{2}\right)$ per $n_{\text{total}}$. The batch size is 4, epoch is 100, and the optimizer is Adam (Kingma & Ba, 2014) whose learning rate is $5 \times 10^{-4}$.

## E. ADDITIONAL COMPLETION TASK: MNIST

We also conduct the image completion task using rotated MNIST. It is thought that (1) original MNIST contains various data symmetries including translation, scaling, and rotation, and (2) we cannot specify them precisely. Figure 11 shows the actual images from the original MNIST datasets. We can confirm that yet we did not conduct any transformation, the images have been already rotated.

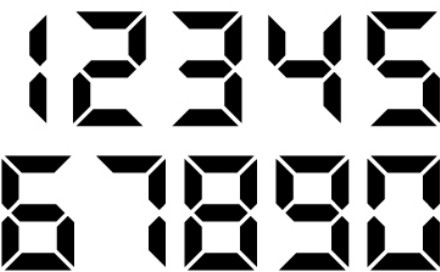

Figure 10: The original data that is used for 2D image-completion task.

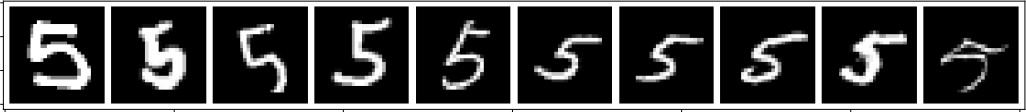

Figure 11: Actual images from original MNIST.

Moreover, factors other than symmetry such as personal habit exist. Indeed, the original MNIST is not good for verify the effectiveness of EquivCNP.

The result is depicted in Figure 12. During this experiment, the batch size is 16, epoch is 30, and the optimizer is Adam whose leraning rate is $5 \times 10^{-4}$. As a result, the model misses the completion when the number of context points is quite a few. On the other hand, when the number of context points is sufficient, the completion results seem well except the $SO(2)$-equivariant model.

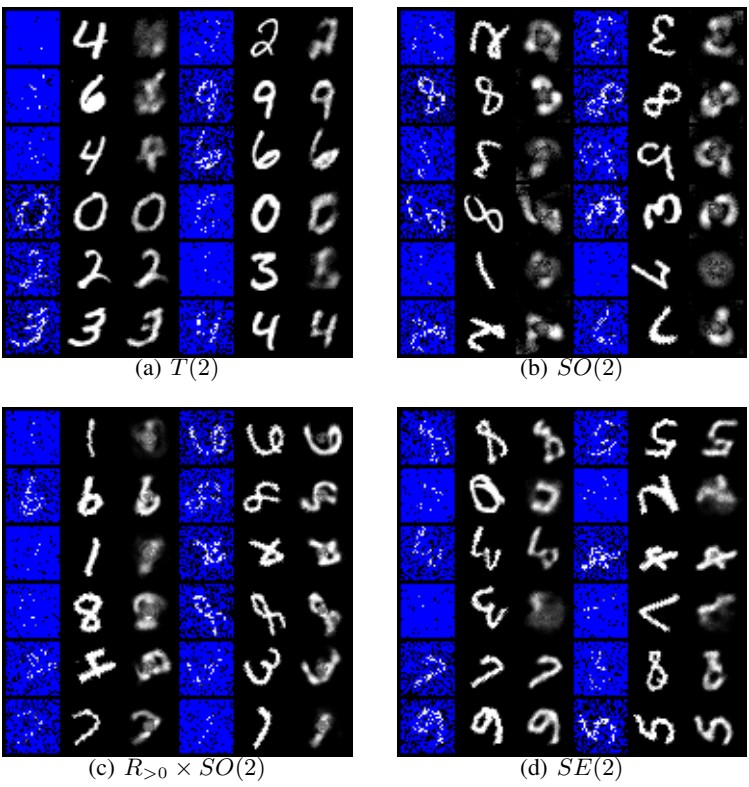

Figure 12: Image-completion task results using rotated-MNIST. In each image, the 1st and 4th columns show context pixels, the 2nd and 5th columns show ground truth images, and the 3rd and 6th columns show completion results. As a result, the model misses the completion when the number of context points is quite a few. On the other hand, when the number of context points is sufficient, the completion results seem well except the $SO(2)$-equivariant model.

