# OpenReview forum: "Group Equivariant Conditional Neural Processes"
_ICLR.cc/2021/Conference — ICLR 2021 Poster_

### Official Review · AnonReviewer3 · 2020-10-27
**Extension of conditional NPs to Lie groups, however concerns remain regarding scalability to other transformations**

**Rating:** 5
**Confidence:** 4

**Review:**

The paper provides an extension of convolution conditional neural processes CNPs to more general Lie group equivariant CNPs. The development of the theory seems sufficiently clear to someone more familiar with the field. However, for newer readers, it seems important to be familiar with background concepts and prior work. This is not a penalizing point but rather just an observation.


The major concern with the body of work is the fact that the experiments seem lacking in application to more realistic data and scenarios. For instance, group invariance is not studied in the context of image classification which would be a direct application where transformations need to be accounted for. Though the image completion task seems interesting, more complex transformations are not handled which leaves this work somewhat incomplete. Indeed, the authors agree to this point in the discussion para.  How would this method scale to other more complex transformation groups such as illumination, object 3D pose etc? Although it might seem easy to brush off these concerns as future work, given that there has been a lot of work in these areas, research that can address these other more challenging yet real-world practical cases need to be addressed atleast in part in a bosy of work.

---

> ### Author Response · Authors · 2020-11-23
> **To Reviewer 3**
>
> Thank you for the thoughtful and constructive review regarding our manuscript, group equivariant neural processes.
>
> This paper aims to verify that we can construct neural processes which has group-equivariance guided by theory, and Adopting EquivCNP to the practical application would be the scope of the next study.
>
> As the reviewer mentioned above, while there are many studies that tackle the image classification tasks by considering various group symmetries, we should tackle meta-(image) classification tasks [1] since our work is located in the meta-learning framework.
> However, it is difficult to tackle the meta-classification tasks due to the neural process architecture; it cannot change the output dimension (i.e. the number of classes for classification) such as MAML[2].
> Note that while there is a work using neural processes to tackle the meta-classification task[3], the architecture of it does not resemble that of conventional neural processes.
> we cannot adopt this architecture to our work straightforwardly.
>
> Scaling EquivCNP to other more complex transformation groups such as illumination, object 3D pose is a very important task in realistic scenarios as the reviewer pointed.
> However, our EquivCNP needs to address the following issue; the computational complexity of 1) LieConv and 2) discretizing E(Z). We will continue to improve the complexity in future work.
>
> - [1] https://arxiv.org/abs/1903.03096
> - [2] https://arxiv.org/abs/2007.02933
> - [3] https://arxiv.org/abs/1906.07697

---

### Official Review · AnonReviewer1 · 2020-10-27
**This paper proposes a member of the neural process family. The improvement or change on CNP is not significant, but the new group equivariance ability is definitely a new and useful feature of NP.**

**Rating:** 7
**Confidence:** 2

**Review:**

This paper proposes a member of the neural process family. To empower the Conditional NP with group equivariance ability, the idea is to replace the original encoder part of Conditional NP with a group convolution-based encoder, which is similar to ConvCNP (Gordon et al. 2019). I am not familiar with the group convolution part but the neural process. It looks LieConv is used as the main group convolution tool but such tool was proposed in another paper (Finzi et al, 2020). The authors did not highlight the contribution made based on LieConv. The improvement or change on CNP is not significant, but the new group equivariance ability is definitely a new and useful feature of NP. The experiment results verified the proposed idea. This paper is the first to introduce the group convolution to NP to my best knowledge. One minor suggestion is that Algorithm 1 is all about group convolution without any computation or inference of the NP part. Considering the title of the algorithm, it would be better to include some details of NP or change the title.

---

> ### Author Response · Authors · 2020-11-23
> **To Reviewer 1**
>
> Thank you for the thoughtful review of our paper.
>
> As the reviewer’s comment, this paper is the first trial to induce the group equivariance to NP, and the most contribution of the paper is we design a theory-guided architecture.
> Furthermore, the reason we used LieConv is well-suitable for EquivCNP at the moment.
> Still, it is also possible to exchange the LieConv to other convolution that can treat group symmetry with much less computational complexity.
> Hence, we will explore the other alternative operation to improve the efficiency of EquivCNP.
>
> > One minor suggestion is that Algorithm 1 is all about group convolution without any computation or inference of the NP part. Considering the title of the algorithm, it would be better to include some details of NP or change the title.
>
> Thank you for your suggestion. Algorithm 1 shows feedforward pass of EquivCNP that predicting $y_t$ corresponding to $x_t$ given context points $(x_c, y_c)$.
> We added the comment in Algorithm 1 so that readers who are familiar with CNP/NP understand it easily.
> Following the reviewer’s suggestion, we modified the title of the algorithm as “Prediction of Group Equivariant Conditional Neural Process”.

---

### Official Review · AnonReviewer2 · 2020-10-27

**Rating:** 4
**Confidence:** 2

**Review:**

### 1. Summary
This paper presents EquivCNP which extends Conditional Neural Processes (CNP) to incorporate symmetries of the data, e.g. rotation and scaling. The approach utilizes a combination of LieConv (Finzi et al., 2020)  and DeepSet (Zaheer et al., 2017) to achieve the equivariance in the data space and permutation invariance across the samples in a dataset. They provide empirical results on a 1D regression task with synthetic and 2D image completion tasks using digital clock digits dataset which they constructed.

### 2. Decision
I am recommending a weak reject for this paper. First, the experimental section does not sufficiently demonstrate the efficacy of the proposed approach. Second, I find the organization of the paper hard to follow at times.
### 3. Supporting arguments
1. The paper’s motivation is interesting and intuitive. Just as ConvCNP (Gordon et al., 2019) extended CNP with translation equivariance, this work proposes to extend CNP with group equivariance. It is generally difficult to build equivariance properties into deep-nets just in the data space alone, CNP requires the additional permutation invariance. Hence, I think the work is of interest to the community.

2. The main weakness of this paper is the experimental section. For example, ConvCNP (Gordon et al., 2019) from last year’s ICLR evaluated on the following datasets: a 1D synthetic dataset, PLastiCC dataset, MNIST, SVHN, CelebA32, CelebA64, ZSMM. On the other hand, this work conducted experiments on a 1D synthetic time series data, and a “digital clock dataset” where the template is shared across train and test sets.

3. The author explained MNIST is “not suitable for evaluating EquivCNP with group equivariance: translation, scaling, and rotation”. I am not convinced by this argument; I think the paper can be strengthened if there is an experiment demonstrating that on a real dataset EquivCNP is beneficial.

4. Furthermore, the image completion results do not compare to any prior benchmark. I don’t see a reason for not comparing.
I think the organization makes the paper difficult to read at times. For example, on page 5, they mentioned $\rho,  \phi, \psi$, which are defined back on page 2. Maybe re-organize such that the modeled conditional distribution, in Sec 4.3,  is at the top of Sec 4; similar to Gordon et al., (2019)’s paper organization.

### 4. Additional feedbacks
- Adding equation numbers and referring to them in the text may help the readers to understand the paper better.

---

> ### Author Response · Authors · 2020-11-23
> **To Reviewer 2**
>
> Thank you for the thoughtful review and constructive feedback regarding our manuscript, group equivariant neural processes.
>
> > The main weakness of this paper is the experimental section. For example, ConvCNP (Gordon et al., 2019) from last year’s ICLR evaluated on the following datasets: a 1D synthetic dataset, PLastiCC dataset, MNIST, SVHN, CelebA32, CelebA64, ZSMM. On the other hand, this work conducted experiments on a 1D synthetic time series data, and a “digital clock dataset” where the template is shared across train and test sets.
>
> - While it is important to see how effective our EquivCNP is in a realistic scenario, the purpose of the experiment in the paper is to verify that our EquivCNP which intends to have group equivariance introduced by the theoretical guide works properly. As an experimental result, we show that our EquivCNP has group equivariance properly and we will focus on exploiting EquivCNP for various applications in future work.
> For example, if we want to exploit EquivCNP to meta-classification tasks, we need to explore another theory-guided design because the current EquivCNP architecture is not suitable for tackling meta-classification tasks; the output  dimension (i.e. the number of classes)  cannot be changed during the training and evaluating.
>
> > The author explained MNIST is “not suitable for evaluating EquivCNP with group equivariance: translation, scaling, and rotation”. I am not convinced by this argument; I think the paper can be strengthened if there is anh experiment demonstrating that on a real dataset EquivCNP is beneficial.
>
> - We are sorry that we didn’t explain reasonably.
> We conducted the image completion task with rotated MNIST, but we could not find the difference between the translation-equivariant model (i.e ConvCNP) and other group equivariant models.
> It is thought that (1) original MNIST contains various data symmetries including translation, scaling, and rotation, and (2) we cannot specify them precisely.
> With actual original MNIST, we can confirm that yet we did not conduct any transformation, the images have been already rotated.
> Moreover, factors other than symmetry such as personal habit exist.
> We revised the corresponding part and added some discussion, and the rotated MNIST results in Appendix E.
>
> > Furthermore, the image completion results do not compare to any prior benchmark. I don’t see a reason for not comparing.
>
> - Following the result of the first experiment, the 1d regression task, EquivCNP with $T(2)$ is equivalent to the prior benchmark (i.e. ConvCNP). Therefore, we can regard the result of the T(2) model in the image completion task as the prior benchmark. To clear it, we modified Figure 4.
>
> > I think the organization makes the paper difficult to read at times. For example, on page 5, they mentioned ρ,ϕ,ψ, which are defined back on page 2. Maybe re-organize such that the modeled conditional distribution, in Sec 4.3, is at the top of Sec 4; similar to Gordon et al., (2019)’s paper organization.
>
> - We reconsidered the organization of our paper and moved part of Section 4.3 to the top of Section 4 following the reviewer 2's suggestion.

---

### Official Review · AnonReviewer4 · 2020-10-28
**Good theory-guided design; okay experiments**

**Rating:** 6
**Confidence:** 3

**Review:**

This paper addresses an important symmetry in meta-learning.  Namely, the context data consists of a set of datapoints in arbitrary order.  The model should thus be permutation equivariant to their order.  At the same time, the data itself may have its own symmetries, e.g. rotation, which the network should likewise be equivariant to.  The authors follow a theory-driven approach, proving in Thm 2 that a function with these two types of symmetries may be factored and represented by a composition of functions reflecting each symmetry individually.  They then design a Neural Process (NP) model, EquivCNP, which reflects this result.  Other works have used permutation equivariance and translation equivariance in NPs, but this is the first to incorporate other symmetry groups.

As a weakness, the method description in Section 4 is often imprecise and unclear as noted in the specific points below.   The method is somewhat novel; it may be described as a combination of two existing methods: permutation-equivariant NPs (Garnelo 18) and LieConv (Finzi 20).  The experiments achieve good results, but could be much more convincing.  They concern fitting only functions on low-dimensional space, report only one metric, and have limited comparison to baselines.  In particular, it does not seem necessary to frame image reconstruction as a meta-learning task.  However, the strength of this paper is the theory-guided model design.  The theorem proved is wholly appropriate to the task and informs the model design very nicely.  In particular, it makes use of the realization that $D_C \to C(X,Y)$ can be $G$-equivariant; that is, the symmetry transformation can be inferred from context and transferred onto the prediction function.  Thus I suggest acceptance (6).

**Specific Point and Questions**
- Sec 2.1 “Weiler & Cesa ... is costly” I do not agree.  However cost is defined this method is on par or better than CNNs
- Sec 2.1 “learning the objective from the data.” I do not follow.  Can you explain further how this applies to EquivCNP?
- Sec 2.1 How do the symmetries in this paper compare to those considered in (Maron et al 2020, On Learning Sets of Symmetric Elements) which also considers permutation and Lie group equivariance together?
- Defn 3, It seems here we are giving all the orbits the same weight? Should there be a measure on orbits?  Considering the $SO(2)$ action on the plane, should not the circular orbits be weighted by circumference?
- Sec 4.2 “LieConv … can handle Lie groups” This is true, but many other architectures such as steerable CNN can as well.
- Sec 4.2 “we define distance in the Lie group.” Since we have lifted from the space $X$, is it important to consider differences between measuring distance on $X$ vs. $G$?
- Sec 4.2 $\| q_i - q_j \|$, how do you define distance between orbits?
- Sec 4.2 Since compact groups have a bi-invariant metric, why not use that instead of $d(u,v)$?
- Page 5, first Eqn, According to which distribution are $(u_i,q_i)$ sampled?
- Page 6, para 1, it is not clear exactly what precisely is meant by discrete or continuous data.
- Page 6 “although we want to convolve $E(Z)$...” I do not follow this sentence.  Can you clarify it?
- Page 7 “MNIST already contains diverse group equivariance” If this is the case then the equivariant model should still have good performance, but perhaps with less improvement over the baseline, correct?
- Page 8, Why not test SO(2)-equivariant model on test data which has only been rotated? Does not the success of $R \times SO(2)$ over the other groups simply show that the others have misspecified (or perhaps underspecified) inductive biases?
- Page 11, second equation, should be $\psi(g^{-1} x, g^{-1} x')$ since it is a left-action.  This will not change the rest of the proof.
- Page 11, proof of (II) implies (I) feels incomplete.  Can you define $\rho, \mathcal{H}$ here?

**Minor Points**
- Sec 3, first equation, $\pi Z_n := \ldots$, the subscripts should be $\pi^{-1}(i)$, not $\pi(i)$. One can see this in a simple example or from the fact $\pi$ is a left-action.
- Sec 3, by $mult(Z’):=$ is sup over $Z_m$ but contains $Z_n$.
- Sec 4.1, “lifting into $K$ elements” How do we know the stabilizer is finite?
- Sec 4.2, note that not all Lie groups are matrix groups as seems to be implied here.
- Sec 4.2, Last Eqn on Page 4, the integral can be written over G since the support of $g_\theta$ limits to the neighborhood of $u$
- Bottom Page 4, footnote, triangle, not trigonometric
- Page 5, first Eqn, sum should be over $(v_j, q_j’)$.
- Since the support of $g_\theta$ is limited to a neighborhood of 1, it may not be necessary to assume $exp$ is surjective since $exp$ is always surjective in a small neighborhood of 1.
- Sec 4.3, first Eqn, the first line is a product of density functions, but the second is a product of distributions.  This is unclear.
- Sec 4.3 EquivDeepSet is not introduced
- Sec 4.3 “a single output per one input” Is this backwards?
- Page 6, line 3, $\phi$ with $\psi$
- Sec 5 “two questions” should “three questions”
- Page 11, Proof switches between $X$ and $S$, make consistent
- Page 11, What do the brackets mean? This notation should be defined.

**Update Based on Author Feedback**
I am grateful to the authors for their detailed replies to my questions. I understand some of the minor points better and am happy they have revised some unclear parts. Overall, I think this paper has some real strengths: theory-guided design, important problem, novel methods. I do feel (similar to R3) that the experiments and applications could have been far more convincing. Weighing these strengths and weakness, I still tilt slightly towards accept (6).

---

> ### Author Response · Authors · 2020-11-23
> **To Reviewer 4 (1/2)**
>
> Thank you for the thoughtful and constructive feedback you provided regarding our manuscript, group equivariant neural processes.
> We corrected all typos and fixed them in the revision paper version.
> We also added the explanation of notation that we did not in the submitted paper.
>
> > Sec 2.1 “Weiler & Cesa ... is costly” I do not agree. However cost is defined this method is on par or better than CNNs
>
> - As the reviewer mentioned, surely steerable CNN is on par or better than conventional CNNs, we trim the corresponding parts.
>
> > Sec 2.1 “learning the objective from the data.” I do not follow. Can you explain further how this applies to EquivCNP?
>
> - We are sorry that it was an unclear description. We revised the corresponding parts as “While both these studies and EquivCNP can handle data symmetries,”.
>
> > Sec 2.1 How do the symmetries in this paper compare to those considered in (Maron et al 2020, On Learning Sets of Symmetric Elements) which also considers permutation and Lie group equivariance together?
>
> - The paper of Maron et al. puts H in a finite group, which is different in aspect from our model dealing with Lie groups.
> Furthermore, even if Maron et al.’s paper were to treat H in the Lie group, the symmetry we are dealing with is local, derived from the exponential map of the Lie group, and Maron et al. deal with the symmetry of the full group.
>
> > Defn 3, It seems here we are giving all the orbits the same weight? Should there be a measure on orbits? Considering the SO(2) action on the plane, should not the circular orbits be weighted by circumference?
>
> - In Definition 3,  $dq’$ is an arbitrary measure on orbits, which is not restricted to a uniform measure. In addition, an appropriate weight can be learned by $g$.
>
> > Sec 4.2 “LieConv … can handle Lie groups” This is true, but many other architectures such as steerable CNN can as well.
>
> - Although other architectures including Steerable CNN can handle specific Lie groups such as translation and rotation, LieConv can treat a wider class of Lie groups in a unified way. That is why we used LieConv.
>
> >Sec 4.2 “we define distance in the Lie group.” Since we have lifted from the space X, is it important to consider differences between measuring distance on X vs. G?
>
>  - Equivariance of convolution is related to group action rather than the topology of $\mathcal{X}$. Thus, we use the left-invariant (pseudo-)distance $d(u,v)$ on a Lie group, which is compatible with group action, although $\mathcal{X}$ may have some distance.
>
> > Sec 4.2 |qi−qj|, how do you define distance between orbits?
>
> - We should have denoted the distance between orbits by $d_{\mathcal{O}}(q_i,q_j)$ instead of $\|q_i - q_j\|$.  $d_{\mathcal{O}}(q_i,q_j)$ is defined by $\inf_{x_i\in q_i, x_j \in q_j} d_{\mathcal{X}}(x_i, x_j)$.
>
> > Sec 4.2 Since compact groups have a bi-invariant metric, why not use that instead of d(u,v)?
>
> - We may use any invariant metric for localization as long as the metric is computable. The reason why we used $d$ in Section 4.2 is it is numerically computable for matrix representation of groups in experiments.
>
> > Page 5, first Eqn, According to which distribution are (ui,qi) sampled?
>
> - $(u_i, q_i)$ is sampling from the measure $d\mu \times dq’$.
>
> > Page 6, para 1, it is not clear exactly what precisely is meant by discrete or continuous data.
>
> - We do not have a precise definition of “discrete” and “continuous” data. However,  we suppose that
> discrete data are drawn from a probability distribution on a finite set or  a countable set with the discrete topology
> and that continuous data on a Euclidean space are drawn from a probability distribution which is absolutely continuous with respect to the Lebesgue measure on the Euclidean space.
>
> >Page 6 “although we want to convolve E(Z) ...” I do not follow this sentence. Can you clarify it?
>
> - Although EquivCNP requires to compute the convolution of $E(Z)$, it cannot be computed in computers as it is since $E(Z)$ itself is a functional representation.
> Hence,  we need to discretize $E(Z)$ so that we can compute in computers.
>
> > Page 7 “MNIST already contains diverse group equivariance” If this is the case then the equivariant model should still have good performance, but perhaps with less improvement over the baseline, correct?
>
> - We are sorry that our explanation confused you.
> As the reviewer mentioned, we could not find a significant difference between the baseline and our model.
> We modified the corresponding part of the explanation and added the experiment result of MNIST image completion to Appendix E.

---

> ### Author Response · Authors · 2020-11-23
> **To Reviewer 4 (2/2)**
>
> > Page 8, Why not test SO(2)-equivariant model on test data which has only been rotated? Does not the success of R×SO(2) over the other groups simply show that the others have misspecified (or perhaps underspecified) inductive biases?
>
> - The images represented as “x1.0” are not scaled but rotated in Figure 4. Indeed, we have tested the $SO(2)$-equivariant model on only rotated test data.
> Also, as the reviewer mentioned, the results depicted in Figure 4 show that group equivariance except for $R\times SO(2)$ is misspecified for the image completion task.
>
> > Page 11, proof of (II) implies (I) feels incomplete. Can you define ρ,H here?
>
> - We are sorry for the incomplete proof. We completed proof of that (II) implies (I). The proof in page 11 is the proof of Theorem 2, and $\rho$ and $\mathcal{H}$ are introduced in the statement of Theorem 2.
>
> > Sec 4.1, “lifting into K elements” How do we know the stabilizer is finite?
>
> - We do not assume that the stabilizer is finite. $K$ is a hyperparameter and we randomly pick $K$ elements $\\{ u_{ik} \\}_{k=1}^K$ in the orbit corresponding to $x_i$.
>
> > Sec 4.2, note that not all Lie groups are matrix groups as seems to be implied here.
>
> - We know of an example that cannot be embedded in a matrix group, such as an infinite set of permutations, but basically we are dealing with geometric operations such as rotation and translation and so on, and so we assume that we have a finite dimensional representation. We will add this assumption in the paper.
>
> > Sec 4.2, Last Eqn on Page 4, the integral can be written over G since the support of gθ limits to the neighborhood of u
>
> - Although the reviewer is mathematically right, in section 4.2, we aim to derive a computable form of convolution. Thus, we explicitly localized and discretized  the support of the measure.
>
> > Since the support of gθ is limited to a neighborhood of 1, it may not be necessary to assume exp is surjective since exp is always surjective in a small neighborhood of 1.
>
> - We removed the condition that exp is subjective.
>
> > Sec 4.3, first Eqn, the first line is a product of density functions, but the second is a product of distributions. This is unclear.
>
> - $\mathcal{N}$ denotes the density function of a normal distribution.
>
> > Sec 4.3 EquivDeepSet is not introduced
>
> - We named $\Phi$ as EquivDeepSet in Section 3.
>
> > Sec 4.3 “a single output per one input” Is this backwards?
>
> - We replaced the sentence “most real-world data have a single output per one input location” by “two data points $(x_i,y_i)$ and $(x_j,y_j)$ within real-world data $\mathcal{D}_C$ have different inputs $x_i\ne x_j$ in most cases, ”
>
> > Page 11, What do the brackets mean? This notation should be defined.
>
> - $\langle \cdot, \cdot \rangle$ is the inner product in the RKHS $\mathcal{H}$. $[\mathcal{Z}]=\mathcal{Z}/S_n$ is a quotient space of $\mathcal{Z}$ by the action of the permutation group $S_n$.

---

### Decision · Program_Chairs · 2021-01-07
**Final Decision**

**Decision:**

Accept (Poster)

**Comment:**

This paper provides a natural combination of conditional neural processes with LieConv models. It is a good step forward for stochastic processes with equivariances. While there is still room to improve the experiments, the authors provided a good response to reviewers, and the paper is a nice contribution.